# Live Triple Gene-Deleted Pseudorabies Virus-Vectored Subunit PCV2b and CSFV Vaccine Undergoes an Abortive Replication Cycle in the TG Neurons following Latency Reactivation

**DOI:** 10.3390/v15020473

**Published:** 2023-02-08

**Authors:** Selvaraj Pavulraj, Rhett W. Stout, Daniel B. Paulsen, Shafiqul I. Chowdhury

**Affiliations:** Department of Pathobiological Sciences and Louisiana Animal Disease Diagnostic Laboratory, School of Veterinary Medicine, Louisiana State University, Baton Rouge, LA 70803, USA

**Keywords:** pseudorabies virus, vectored vaccine, latency reactivation, thymidine kinase, glycoprotein E, triple mutant, trigeminal ganglion, pig, PRVtmv vector, DIVA

## Abstract

Like other alpha herpesviruses, pseudorabies virus (PRV) establishes lifelong latency in trigeminal ganglionic (TG) neurons. Upon stress, the latent viruses in the TG neurons reactivate and are transported anterograde from the neuron cell bodies to the nerve endings in the nasal mucosa, where they replicate and are discharged in the nasal and oral secretions. Consequently, the virus is transmitted to other naïve animals. This cycle of latency and reactivation continues until the animal dies or is slaughtered. We have constructed a PRV triple mutant virus (PRVtmv) and used it as a live subunit vaccine vector against porcine circovirus 2b (PCV2b) and classical swine fever virus (CSFV) (PRVtmv+). We compared the latency reactivation properties of PRVtmv+ with its parent wild-type (wt) Becker strain following intranasal infection. The results showed that PRV wt and PRVtmv+ established latency in the TG neurons. Based on nasal virus shedding, immediate early (infected cell protein 0; ICP0) and late genes, MCP (major capsid protein) and gC (glycoprotein C) transcriptions, and viral DNA copy numbers in the TGs of latently infected and dexamethasone (Dex)-treated pigs, both PRV wt and PRVtmv+ reactivated from latency. We noticed that PRV wt virus replicated productively in the terminally differentiated, postmitotic TG neurons, but PRVtmv+ failed to replicate and, therefore, there was no virus production in the TG. In addition, we found that only the PRV wt virus was shed in the nasal secretions following the Dex-induced reactivation. Our results demonstrated that the PRVtmv+ is safe as a live viral subunit vaccine vector without the possibility of productive replication in the TG upon reactivation from latency and without subsequent nasal virus shedding. This property of PRVtmv+ precludes the possibility of vaccine virus circulation in pigs and the risk of reversion to virulence.

## 1. Introduction

Pseudorabies, or Aujeszky’s disease (AD), caused by the pseudorabies virus (PRV), is an economically significant viral disease of pigs [1,2]. Following oronasal transmission, PRV initially replicates in the nasal epithelial cells. After that, the virus enters the nerve endings of the trigeminal ganglionic (TG) neurons, transported to the cell bodies, where it establishes life-long latency [3]. During latency, the virus does not replicate. However, there is latency-associated transcript (LAT) gene transcription [4]. Upon stress and/or dexamethasone administration, the latent virus reactivates, replicates in the TG neurons, and is transported retrogradely to the nasal mucosa where it replicates and sheds [5]. Consequently, the virus can be transmitted to uninfected animals in the herd and maintained in the swine population. By using the glycoprotein E (gE) gene-deleted marker PRV DIVA vaccine (distinguishing infected from vaccinated animals), PRV has been eradicated from the domestic pig population in the majority of European Union (EU) countries and North America [6,7,8]. However, PRV is endemic in domestic pigs in China and many Asian and African countries [9]. Importantly, PRV is also endemic in wild boars and feral pigs worldwide, including EU countries and the U.S. Thus, there is a constant risk of spillover of PRV infection from wild pig populations to domestic animals [10].

We recently constructed a PRV triple mutant virus (PRVtmv) by deleting the genes coding for envelope proteins gE and gG, and the non-structural protein thymidine kinase (TK) [1]. Both gE and gG are not required for virus replication and retrograde axonal transport, but gE is required for axonal anterograde transport [11,12]. The gG is a viral chemokine binding protein, allowing the virus to evade the initial immune response by inhibiting the migration of immune cells (neutrophils, lymphocytes, and monocytes) [13]. The viral TK is not essential for the replication in the epithelial cells both in vitro and in vivo, because the cellular TK complements the viral TK. However, the viral TK is critical for virus replication in the terminally differentiated postmitotic neurons lacking the cellular TK [6,14]. We used PRVtmv as a subunit vaccine vector for porcine circovirus type 2b (PCV2b) and classical swine fever virus (CSFV). Specifically, chimeric PCV2b-capsid protein (Cap) was inserted in the TK deletion locus. Additionally, chimeric CSFV envelope proteins E2 and Erns fused with porcine granulocytic macrophage colony-stimulating factor (Erns-GM-CSF) were inserted into the gE and gG deletion loci, respectively. The resulting PRVtmv+ (PCV2b and CSFV subunit vaccine vector) is efficacious against the PCV2b challenge in pigs [1]. PRVtmv+ also produced neutralizing antibodies against CSFV [1], and preliminary PRVtmv+ vaccination CSFV challenge results revealed that it is also protective against CSFV [15]. 

In the present study, our goal was to investigate whether the PRVtmv+ established latency in the TG neurons, reactivated upon dexamethasone (Dex)-induced reactivation and shed in the nasal secretions. The results demonstrated that both PRV wt and PRVtmv+ established latency in the TG neurons and reactivated upon Dex treatments. We noticed that only the wt virus replicated in the TG neurons, transported to the nerve endings in the nasal mucosa, and shed in the nasal secretions. 

## 2. Materials and Methods

### 2.1. Ethical Statement

The study was performed in accordance with protocols approved by the LSU Institutional Animal Care and Use Committee, Louisiana State University (IACUC Protocol #20-027; dated 4 January 2020).

### 2.2. Cells and Medium

Swine kidney (SK; #CRL-2842, ATCC^®^, Manassas, VA, USA) and Madin Darby bovine kidney cells (MDBK; #CCL-22, ATCC^®^) were propagated in Dulbecco’s modified eagle medium (DMEM; #10-017-CV, Corning^®^, Corning, NY, USA) supplemented with 10% heat-inactivated fetal bovine serum (FBS; Equa1FETAL, Atlas Biologicals, Fort Collins, CO, USA) and 1 × antibiotic-antimycotic solution [34-004-CI, Corning^®^] (growth medium). 

### 2.3. Viruses

PRV wt Becker strain is a virulent field isolate from a dog at Iowa State University [16]. PRVtmv+ vaccine virus was constructed and characterized earlier [1]. Both PRV wt and PRVtmv+ were propagated in SK cells. Low-passage virus stocks were titrated by plaque assay in MDBK cells as described previously [17], aliquoted, and maintained at −80 °C.

### 2.4. Animals and Experimental Design

Animal handling, sample collection, virus infection, dexamethasone-induced latent virus reactivation, and euthanasia protocols were approved by the LSU Institutional Animal care and Use Committee (Protocol #20-027). Five 8-week-old healthy Yorkshire pigs were purchased from a PRV- and PCV2b-free supplier (Valley Brook Research, Madison, GA, USA). Before inclusion in the study, pigs were tested for bovine viral diarrhea (BVD)-free status by the serological assay as described previously [17]. After acclimatization for a week, pigs were randomly divided into two groups: 2 pigs in group 1 and 3 pigs in group 2 (Figure 1). The pigs in groups 1 and 2 were housed in separate rooms with negative pressure in BSL-2 animal holding facility at the School of Veterinary Medicine, Louisiana State University. The pigs were housed separately in individual cages. To prevent cross-contamination between the groups, footbaths were located at the entrance of each room, and the PRVtmv+-inoculated group was attended to first before handling the pigs in the wt group. All sanitary precautions were taken to prevent cross-contamination between the groups. All bedding materials and excretions from pigs were sterilized before discarding.

### 2.5. Primary Virus Infection, Dexamethasone-Induced Latent Virus Reactivation, and Clinical Evaluation

Each pig in the PRV wt infection group 1 was intranasally (IN) infected with total 2 × 10^6^ PFUs/nostril (total 4 × 10^6^ PFUs/pig). The pigs in the PRVtmv+ vaccine group were inoculated IN with 4 × 10^6^ PFUs per nostril (total 8 × 10^6^ PFUs/pig) and subcutaneously (S/C) with filtered (0.2 µm pore size) 4 × 10^6^ PFUs. On the day of infection, each pig received Noromycin^®^ 300 LA I/M injection (Norbrook, Lenexa, KS, USA; 20 mg/kg body weight) to prevent secondary bacterial infections. At 28 days post-infection (dpi), pigs in both groups received the dexamethasone (Dex) intravenously (I/V) (0.5 mg/kg), followed by two more S/C Dex injections (0.25 mg/kg) on 29 and 30 dpi. 

Pigs were routinely monitored for obvious clinical illness, feed intake, and water intake. Bodyweight and temperature were recorded. Clinical assessment included coughing, sneezing, nasal discharge, depression, respiratory difficulties, and other systemic illnesses (Table 1).

### 2.6. Clinical Sample Collection and Processing

Nasal swabs were collected in 2 mL of DMEM, supplemented with 3× antibiotic-antimycotic solution and 2% FBS. Nasal swab samples were aliquoted and stored at −80 °C until use. Blood samples collected for sera were processed, aliquoted, and stored at −80 °C.

### 2.7. Euthanasia, Necropsy—Sample Collection and Processing

At five days post-dexamethasone treatment (5 dp-Dex), pigs were euthanized with xylazine and Euthasol^®^ (euthanasia solution; pentobarbital sodium and phenytoin sodium). At necropsy, TGs were collected for histopathology (10% formalin), virus isolation, and qPCR assays (dry ice). Formalin-fixed tissues were paraffinized, sectioned, and processed for histopathology (H&E staining).

### 2.8. Serum Virus Neutralization Assay 

PRV-specific neutralizing antibody titers in sera were determined by a standard plaque reduction assay (50% reduction/neutralization of the approx. 100 PFUs) as described previously [1]. Viral plaques were counted at 72 h post-infection (hpi) after the cells were fixed (4% paraformaldehyde) and stained (0.1% crystal violet). 

### 2.9. DNA/RNA Isolation, cDNA Synthesis, and PRV-Specific Quantitative PCR (q-PCR)

To determine the PRV genome copies in the nasal swabs and in the TG, total DNA was isolated as described previously [1] using the QIAamp^®^ DNA mini kit (#51306, Qiagen, Hilden, North Rhine-Westphalia, Germany). Before DNA isolation from TG, tissues were homogenized using 2.8 mm ceramic beads (#15-340-154, Thermo Fisher Scientific^®^, Waltham, MA, USA) in Pre-cellys 24 homogenizer (#13112, Bertin Instruments, Rockville, MD, USA). To determine the transcription and quantification of the targeted immediate early (infected cell protein 0 - ICP0) and late (major capsid protein - MCP and glycoprotein C - gC) PRV genes in the TG neurons, RNA was isolated. Briefly, after the homogenization of TG, RNA was isolated using an RNeasy mini kit (#74104, Qiagen). DNA contamination in RNA samples was removed by RNase-Free DNase Set (#79254, Qiagen). Finally, cDNA synthesis was performed using the Verso cDNA synthesis kit (AB-1453/A, Thermo Fisher Scientific^®^). 

PRVtmv+ genomic copies were determined by TaqMan probe-based real-time qPCR in ABI PRISM™ 7900HT Sequence Detection System (Applied Biosystems, Waltham, MA, USA), using targeted genes ICP0-, MCP-, and gC-specific primers and probes (Suid herpesvirus 1 strain Becker, GenBank accession # JF797219.1; Table 2). Each time, the PCR reaction setup was run with six standards of known quantity (10^1^ to 10^6^ copies per reaction).

The PRV gene copies were calculated by normalizing the MCP-specific CT values against the standard curve generated based on the CT values obtained for the known housekeeping gene, GAPDH copies (two copies/cell), in the same cells. The mean copies of the PRV-MCP gene per one million cells were then plotted. 

For the PRVtmv+ latency control, we used the TG samples collected and stored at −80 °C from our previous vaccination experiment [1]. To demonstrate that RNA samples treated with the DNAse were free from residual DNA, treated RNA samples without cDNA synthesis were also used as controls. PRV genome copies in TG were normalized to endogenous host-specific swine housekeeping gene, glyceraldehyde 3-phosphate dehydrogenase (GAPDH; GenBank accession #AF017079.1). The assays were performed in duplicate. The qPCR genome copy results are expressed as PRV genome copies per million cells. The RT-qPCR gene transcript copy results are expressed as transcript copies/ng of RNA. 

### 2.10. Statistical Analysis

All data are expressed as means ± standard deviation. Statistical analyses were performed using GraphPad PRISM^®^ software version 5.04 (GraphPad Software, San Diego, CA, USA). The two-way analysis of variance (ANOVA) followed by Bonferroni post-tests to compare replicate means by row were performed. A value of *p* ˂ 0.05 was considered statistically significant. 

## 3. Results

### 3.1. Clinical Evaluation

On 3 dpi, PRV wt-infected pigs started to show moderate fever (40.6 °C) and mild to moderate respiratory signs (clinical score 9) (Table 1; Figure 2A,B; Appendix A). The clinical signs in wt-infected pigs were most severe on 4 dpi, with high fever ranging between 40.8 °C and 41.8 °C and a clinical score of 11.5 characterized by shivering, bilateral nasal discharge, coughing, sneezing, tear staining, and moderate to severe respiratory difficulties (Figure 2A,B and Figure 3A). The pigs were off-feed for a day (4th dpi) and had reduced feed and water intake for two days (3rd and 5th dpi). By 7 dpi, the clinical signs gradually subsided. We noticed that one wt-infected pig developed a unilateral corneal ulcer on the right eye, leading to descemetocele on 7 dpi (Figure 3B), which persisted until the day of euthanasia (33 dpi). In contrast, the PRVtmv+-infected pigs were clinically normal until euthanasia on 33 dpi, including 5 day post-reactivation (Figure 2A,B and Figure 3C).

### 3.2. Nasal Virus Shedding following Intranasal (IN) Administration of PRV wt and PRVtmv+

Following primary intranasal infection of pigs, both PRV wt and PRVtmv+ replicated in the nasal mucosa and shed in nasal secretions. As shown in Table 3, infectious viral plaques were recovered only from the nasal swabs of PRV wt-infected pigs on 3 dpi. Consistent with this result, on 3 dpi and 5 dpi, the corresponding genomic copy numbers in the nasal swabs of PRV wt-infected pigs were also significantly higher (30,000 folds) compared with the PRVtmv+-infected pigs (Table 3). Further, low-level nasal virus shedding (approx. 2 × 10^2^ 5 × 10^2^ genome copies) persisted until 21 dpi in the PRV wt-infected pigs but was cleared by 28 dpi. 

### 3.3. Following Dex-Induced Reactivation, Only the PRV wt-Infected Pigs Shed the Virus in Their Nasal Secretions 

On 28 dpi, nasal virus shedding was no longer detectable by qPCR, indicating that the virus established latency in the TG neurons. Therefore, starting at 28 dpi, we began administering Dex to all five pigs to induce the reactivation of latent PRVtmv+ and PRV wt viruses. Nasal swabs of pigs collected on 3 and 5 dp-Dex were tested by qPCR for virus shedding. The results presented in Figure 4 and Appendix A demonstrate that on 4 dp-Dex, one of the two pigs infected with the wt shed the virus, and on day 5 post-Dex, both the pigs shed the wt virus. However, based on qPCR results (Figure 2; Appendix A), none of the three pigs in the PRVtmv+ group shed detectable virus in the nasal secretions after the Dex treatments.

### 3.4. The PRV wt and PRVtmv+ Established Latency in the TG Neurons and Reactivated following Dex Injection, but Only PRV wt and Not PRVtmv+ Replicated in the TG Neurons

To demonstrate that PRVtmv+ was transported retrogradely to the TG neurons following intranasal infection and established latency in the neuron cell bodies, we determined viral genome copies in TGs of wt and PRVtmv+-infected/vaccinated pigs by qPCR, targeting the PRV MCP gene. Specifically, we used the TGs collected from pigs at 53 days post-PRVtmv+ vaccination/21 days post-PCV2b challenge from a recent PRVtmv+ vaccination– PCV2b challenge study [1]. The results depicted in Figure 5 and Appendix A reveal that a mean of 1643 PRV genome copies per million cells was detected in the PRVtmv+ latently infected TGs. For comparison, we did not have corresponding PRV wt TG samples. Nevertheless, PRV wt latency in the TG neurons was not questionable following primary intranasal infection. To determine whether the latent PRVtmv+ virus replicated in the TG neurons following latency reactivation, we quantified PRVtmv+ genome copy numbers in the TGs of vaccinated pigs at five days post-Dex treatment and compared it with the corresponding PRV wt genome copy numbers in similarly treated PRV wt-infected pigs. The results in Figure 5 demonstrate that at 5 days post-Dex treatment, 122,924 mean genome copies per million cells in the PRV wt-infected TGs were detected, while the corresponding mean genome copy numbers in similarly treated PRVtmv+-infected TGs were 1174/million cells, which is 100-fold less than the numbers obtained from the TGs of wt-infected pigs. Remarkably, the comparison of genome copy numbers obtained from the TGs of PRVtmv+-vaccinated pigs either during latency or at five days post-Dex treatment (latency reactivation) were similar. These results suggest that PRVtmv+ did not replicate in the TG after the Dex treatment, while PRV wt replicated.

To confirm that latent PRVtmv+ reactivated in the TG neurons following the Dex treatments, we investigated the transcription of the viral immediate early gene, ICP0, during latency (of PRVtmv+-infected, negative control) and at 5 days post-Dex treatment (of PRVtmv+ and PRV wt-infected pigs). The results depicted in Figure 6A and Appendix A demonstrate that regardless of PRV wt or PRV vaccine vector-infected pigs, there was ICP0 transcription in the TGs following the Dex treatment. We noticed that the ICP0 copy numbers for the PRVtmv+-infected pigs were slightly lower than those of PRV wt.

The postmitotic neuron does not have a complementary TK enzyme as in the nasal epithelium. Therefore, we expected that PRVtmv+ lacking the viral TK could not replicate in the TG neurons. To prove that PRVtmv+ did not replicate, we investigated transcription levels of the DNA replication-dependent late γ genes, MCP and gC, in the TG neurons during latency (of PRVtmv+-infected pigs) and at 5 days post-Dex treatment (of PRVtmv+ and PRV wt-infected pigs). The rationale was that detecting the MCP and gC transcriptions, or no transcriptions, following the Dex treatment would indirectly indicate that the virus replicated or failed to replicate (abortive replication), respectively. The results depicted in Figure 6B,C, and Appendix A reveal that MCP (Figure 6B) and gC (Figure 6C) gene transcriptions occurred only in the TGs of PRV wt-infected pigs and not in the PRVtmv+-infected pigs. These results proved that both PRV wt and PRVtmv+ viruses reactivated in the TG neurons upon Dex treatments. However, only the PRV wt virus replicated in the TG.

### 3.5. Following Dexamethasone (Dex)-Induced Latency Reactivation, Only PRV wt but Not PRVtmv+-Inoculated Pigs Had a Memory Serum Virus Neutralizing (SN) Antibody Response 

Following latency reactivation in the TG, PRV and other alpha herpesviruses replicate initially in the TG neurons at a low level. Subsequently, progeny viruses travel anterogradely from neuron cell bodies down the axon to nerve endings in the nasal mucosa and replicate there, resulting in nasal virus shedding. Since the pigs were pre-exposed to PRV, there was a B cell response and generation of memory B cells. Consequently, upon virus reactivation in the TG and replication in the nasal epithelium, memory B cells are expected to undergo a recall immune response resulting in a rapid rise in SN antibody titers. Therefore, a drastic increase in the PRV-specific SN antibody titers due to memory immune response following Dex-induced latency reactivation is indirect evidence of latent virus reactivation in the TG, followed by virus replication in the nasal epithelium, and completion of all the steps in between noted above. As shown in Figure 7 and Appendix A, in the case of PRVtmv+-vaccinated/infected pigs, the average SN titers peaked at 50 on 15 dpi, whereas in PRV wt-infected pigs, the peak SN titer rose to 235 at 21 dpi. Subsequently, SN antibody titers dropped to 15 and 120 in PRVtmv+- and PRV wt-infected pigs at 28 days when the pigs stopped shedding the virus and most likely established latency. On 5 dp-Dex, the mean PRV-specific SN antibody titers rose more than ninefold compared to 0 dp-Dex (from 120 to 1096) in PRV wt-infected pigs. In contrast, the corresponding serum virus neutralization titer in PRVtmv+-infected pigs dropped from 15.1 to 11.7 (Figure 7; Appendix A). These results are consistent with the finding that only PRV wt is shed in the nasal secretions following Dex-induced latency reactivation (Figure 4).

## 4. Discussion

Previously, we characterized the PRVtmv+ subunit vaccine vector against PCV2b and CSFV. In PRVtmv+, three viral genes, namely TK, gG, and gE, were deleted and replaced with three chimeric genes, PCV2b Cap, CSFV Erns-GM-CSF, and CSFV E2, respectively [1]. In cell cultures, PRVtmv+ replicated with similar kinetics and yield to the wild-type, but has a small plaque phenotype. PRVtmv+ also replicated in the nasal epithelium of pigs but for a shorter time, 4 dpi versus 15–20 for wt (this study), and was utterly attenuated [1]. In this study, we demonstrated that following intranasal infection, PRV wt and PRVtmv+ established latency and reactivated in the TG neurons upon Dex treatment. However, only PRV wt virus and not PRVtmv+ replicated in the TG neurons and shed in the nasal secretions upon Dex-induced reactivation. Consistent with this finding, only the PRV wt-infected pigs had memory B cell response and had a significant rise in PRV-specific serum neutralizing antibody upon reactivation from latency. 

It is well-established and documented knowledge that all animal alpha herpesviruses establish latency in the sensory neurons of the trigeminal ganglia upon primary intranasal infection. Further, it is also well known that envelope proteins gE, gG, and TK genes are not necessary for virus entry into the nerve termini of the trigeminal nerve and retrograde transport to the cell bodies in the TG neurons where both wt and PRVtmv+ (gE, gG, and TK-deleted) viruses establish latency [18,19,20,21,22]. In vitro, in cell culture, incoming linear viral genome circularizes immediately upon entry into the nucleus before a well-regulated gene transcription/translation scheme begins. Immediate early and early genes are transcribed/translated first before the genome replication and late gene transcription/translation. During latency, the viral genome assumes a circular configuration and resides in the nucleus of the neuron.

Further, except for latency-associated transcript (LAT), there is no other gene transcription or translation and no productive virus replication during latency. Upon stress or dexamethasone-induced reactivation of the latent herpesvirus, the latent virus reactivates and sheds in the nasal secretion [23]. In the TG neurons, viral gene transcription is also similarly regulated as in cell culture —the immediate early gene transcription occurs first from the latent circular genome upon reactivation. This is followed by early gene transcription and translation, which includes virally coded TK, required for the virus replication. Lastly, late proteins are produced, including capsid and envelope proteins required for virus assembly. Notably, many, if not all, late protein transcriptions and translations are dependent on DNA replication. 

We designed the PRVtmv with the following hypotheses: (i) it can be used as a live subunit vaccine vector for immunization against other swine viruses because the virus will replicate in the nasal epithelium and induce mucosal, humoral, and cellular immune responses; (ii) the intranasal vaccination route with a live virus vector would be less affected by pre-existing immune interference; (iii) after primary infection of the nasal epithelium, the virus would enter the nerve endings of the trigeminal nerve and be transported to the cell bodies in the TG neurons; (iv) the vaccine virus will establish latency in the TG neurons, but under no circumstances can the virus be shed in the nasal secretions, and therefore, the vaccine virus will not be transmitted to other pigs or maintained in the pig population; and (v) finally, the vaccinated animals can be distinguished from the PRV wt-infected pigs based on the DIVA (distinguishable from the field wild-type PRV) property, which will be critical if the vaccine needs to be used in PRV-free countries. We have already characterized PRVtmv+ in vitro and in vivo in pigs and determined its vaccine efficacy against PCV2b [1] and CSFV [15].

Our experimental design for this study was based on the following dogmas. (i) Neurons are postmitotic and, as such, do not code for TK enzyme; therefore, the TK minus viruses do not replicate in the neurons [24]. (ii) TK minus animal alpha herpesviruses replicate in vitro in nonneuronal epithelial cells and in vivo in nasal or other epithelial cells, because the cellular TK in these cells can complement the viral TK; therefore, they can be readily propagated in cell cultures, the stock viruses can be prepared, and they are suitable as a live vaccine. (iii) Alpha herpesvirus anterograde axonal neuronal transport from the neuronal cell body to the axon termini is mediated by gE and Us9, and axonal neuronal transport of infectious fully enveloped PRV occurs within transport vesicles along the axon [25,26]. Therefore, virus replication and assembly in the TG neurons must first be completed before the virus can be transported to the nasal epithelium and shed in the nasal secretions. We hypothesized that regardless of PRV wt or PRVtmv+, both viruses would reactivate upon Dex and transcribe the immediate early gene ICP0; however, only the PRV wt and not the PRVtmv+ would transcribe the early protein TK, required for virus replication in the TG neurons. As a result, the PRV wt—and not the PRVtmv+—can replicate in TG neurons, and it forms the enveloped virus particles for vesicular anterograde transport to the nerve termini in the nasal epithelium and the nasal virus shedding. 

Based on the above discussion, we never questioned the establishment of latency in the TGs of both PRV wt and PRVtmv+ after primary intranasal infection. Moreover, for PRV wt, Dex-induced latent virus reactivation and nasal virus shedding were expected. Thus, we used only two pigs as a positive control for PRV wt. For PRVtmv+, we expected the virus not to replicate in the TG neurons, even if the latent virus was reactivated. To prove that the PRVtmv+ reactivated in the TGs of Dex-treated pigs, we tested the transcription of the immediate early gene, ICP0, which would be transcribed immediately after reactivation regardless of whether the virus expressed TK or not. Our results demonstrate the ICP0 transcription in the TGs of all three Dex-treated pigs. To prove that PRVtmv+ failed to replicate in the same TG neurons, we compared the number of PRVtmv+ genomic copies between the TGs of latently infected- and Dex-treated pigs. As expected, the genomic copy numbers in both cases were similar. The final proof that PRVtmv+ did not replicate in the TG neurons of Dex-treated pigs was that the DNA replication-dependent gamma 2 (late gene) gC transcription was not detectable in all three pigs. Additionally, the leaky late (partially DNA replication-dependent), major capsid protein VP5 gene was not transcribed. On the contrary, both gC and VP5 transcripts were readily detectable in the TGs of Dex-treated wt-infected pigs (positive control). 

We have confirmed in two ways that after Dex-induced latency reactivation of PRVtmv+, there was no nasal virus shedding because PRVtmv+ replication in the TG neurons was aborted. Our data revealed that PRVtmv DNA was present in the TG neurons of latently infected pigs. There was ICP0 gene transcription in the TG neurons upon Dex-induced reactivation. Still, there were no transcriptions of the DNA replication-dependent, late viral genes gC and MCP in the TG neurons [27]. Second, we showed that only the PRV wt DNA was detected in the nasal swabs of PRV wt-infected pigs by 4–5 days of Dex treatment, but no PRVtmv+ DNA was demonstrable in nasal swabs of PRVtmv+-infected pigs. Further, we showed that while PRV-specific SN antibody titers increased 9-fold (approx. from 120 to 1096) at 5 dp-Dex in the wt-infected pigs, the corresponding SN titers in PRVtmv+-infected pigs dropped from 15 to 11. 

This rapid increase in SN titers in the PRV wt-infected pigs, despite the Dex-induced immunosuppression, could be explained by the following factors. First, the pigs were pre-exposed to PRV, resulting in the generation of memory B cells. Second, upon wt virus, latency reactivation due to immunosuppression, and subsequent replication in the nasal epithelium, the memory B cells underwent a secondary recall immune response resulting in a rapid memory B cell proliferation, differentiation to plasma cells, and antibody production. Third, the proliferating memory B cells and differentiated plasma cells have decreased susceptibility to dexamethasone-induced apoptosis. These factors correlate well with the rapid rise in neutralizing antibody titers of wt-infected pigs by 5 dp-Dex, a classical recall response [28]. These findings are consistent with the underlining theme that memory immune response only occurs after the reactivated virus replicates in the TG; progeny viruses are then transported down the axons to the nerve termini in the nasal epithelium and subsequently replicate there. 

Previous studies by other investigators revealed that pre-colonization of TG with a latent PRV prevented the latent infection in the TG with a superinfecting PRV strain [29]. Even though we did not validate the latter phenomenon in this study, we believe it would also be applicable for PRVtmv+-infected pigs. Additionally, the PRVtmv+ has the DIVA property [7]. Therefore PRVtmv+ can also be used as a safe subunit vaccine vector in countries where PRV has been eradicated from the domestic pig population. The potential advantages of using PRVtmv for subunit vaccine vectors are as follows: (i) the PRVtmv+ is highly attenuated for pigs, yet replicates well enough in the nasal mucosa; (ii) it is highly immunogenic, thus inducing a protective immune response in pigs; (iii) it has the DIVA property; (iv) it can potentially accommodate large inserts of 10–15 kb; (v) it stably expresses inserted chimeric proteins; (vi) the vector virus establishes latency in trigeminal ganglion and reactivates but fails to replicate in the TG neurons, leading to no nasal virus shedding; and (vii) it is suitable for the construction of polyvalent subunit vaccines.

## Figures and Tables

**Figure 1 viruses-15-00473-f001:**
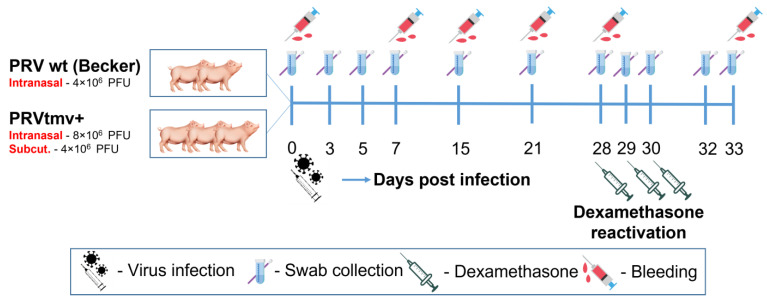
Infection, sample collection, dexamethasone-induced latency reaction, and euthanasia scheme for the animal experiment. PFU—plaque-forming units.

**Figure 2 viruses-15-00473-f002:**
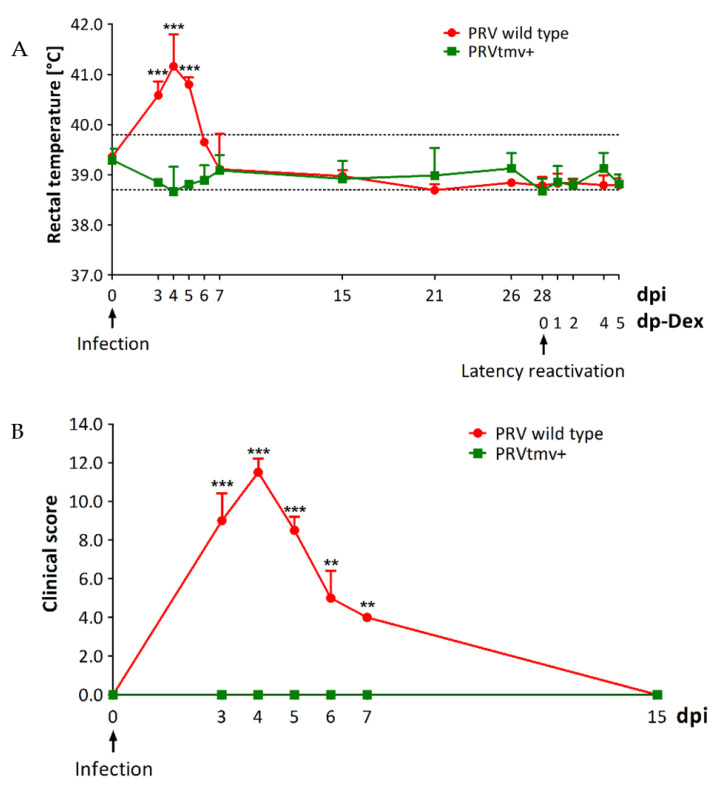
Clinical assessments. (**A**) Rectal temperature of pigs following primary infection and Dex-induced latency reactivation. Shown is the mean temperature of each treatment group with standard deviation (SD). The data represent the mean + SD. There were significant differences in rectal temperature between PRV wild-type (wt) and PRVtmv+ groups. Two-way ANOVA followed by Bonferroni post-tests to compare replicate means by row; *** *p* < 0.001 between PRV wt and PRVtmv+ group. (**B**) Clinical scores in pigs after infection. The scoring criteria are listed in Table 1. Shown is the mean of each group with SD. Two-way ANOVA followed by Bonferroni post-tests to compare replicate means by row; ** *p* < 0.01 and *** *p* < 0.001 between PRV wt and PRVtmv+ groups. dpi—days post-infection; dp-Dex—days post-dexamethasone injection.

**Figure 3 viruses-15-00473-f003:**
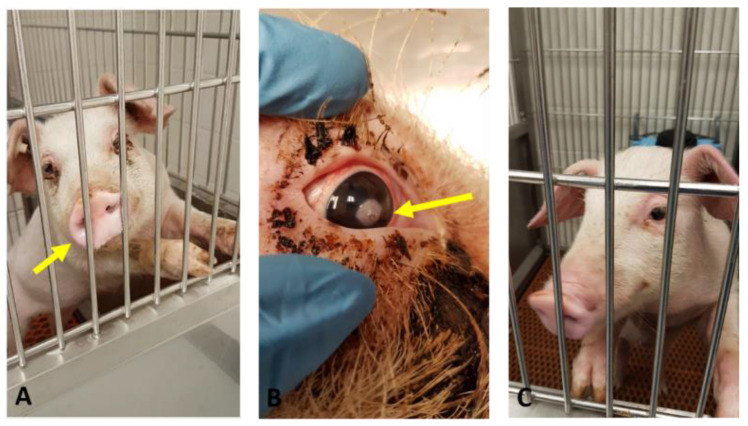
Clinical evaluation of pigs following pseudorabies virus wild-type (PRV wt) or PRVtmv+ vaccine virus infection. (**A**) PRV wt-infected pig showing bilateral nasal discharge and tear staining on 3 dpi. (**B**) Unilateral corneal ulcer and descemetocele on the right eye of PRV wt-infected group on 7 dpi. (**C**) PRVtmv+-infected pigs were healthy without any clinical signs until euthanasia; shown is a pig on 3 dpi.

**Figure 4 viruses-15-00473-f004:**
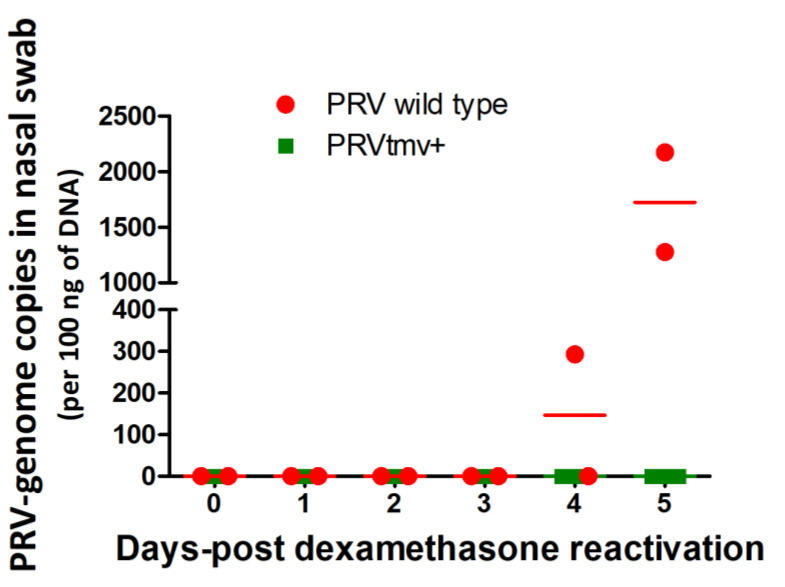
Nasal virus shedding following Dex-induced latency reactivation in PRV wt and PRVtmv+-infected pigs. Nasal swabs collected from pigs daily between 0 and 5 days post-dexamethasone injection (dp-Dex) were tested by qPCR targeting PRV–major capsid protein (MCP) gene. PRV genomic copy numbers were calculated according to the CT values of a standard curve. The mean copy numbers of PRV genome in 100 ng of total DNA from nasal swabs of two independent qPCR analyses of each animal from both groups on 0, 1, 2, 3, 4, and 5 dp-Dex injection. The values represent the mean for each animal in each group.

**Figure 5 viruses-15-00473-f005:**
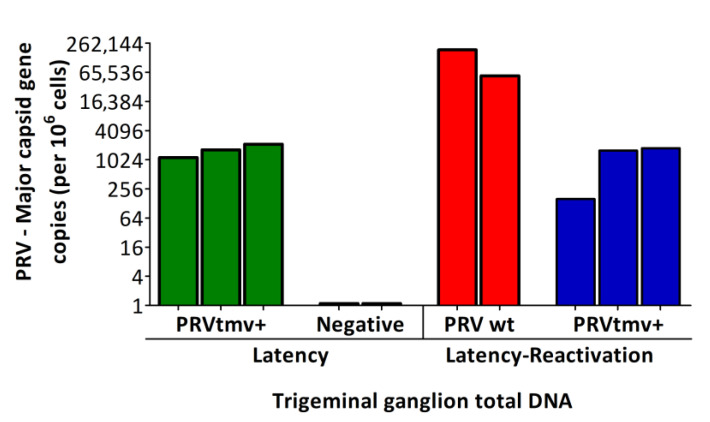
PRV genome copies in the trigeminal ganglion (TG) of infected/immunized pigs determined by PRV major capsid protein (MCP) gene-specific qPCR. Pigs were infected with PRV wt or immunized with PRVtmv+ vaccine. At 28 days post-infection/immunization, latently infected pigs were reactivated by Dex injection. At five days post-reactivation, pigs were euthanized, and TGs were collected and stored frozen (−80 °C). Total DNA was isolated from 25 mg of TG tissues. The qPCR was performed on isolated DNA samples targeting the PRV-MCP gene. The PRV gene copies were calculated by normalization of the MCP-specific CT values against the standard curve generated based on the CT values obtained from the same samples for a cellular housekeeping GAPDH gene. The mean copy numbers of the PRV-MCP gene copies per one million cells are shown. Two independent qPCR tests were performed for each TG sample. The bar graph represents the individual values in each group. PRV wt—PRV wt-infected pigs (*n* = 2); PRVtmv+—PRVtmv+ vaccine-immunized pigs, latency reactivated (*n* = 3); Negative—TGs from mock-infected pigs (stored in −80 °C) from the previous study [1].

**Figure 6 viruses-15-00473-f006:**
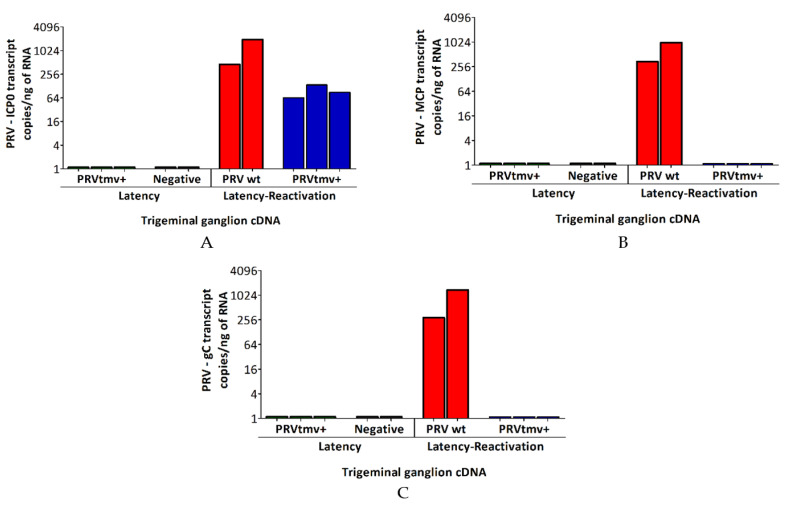
Quantifying targeted, temporally (immediate early and late) expressed PRV gene transcripts in the trigeminal ganglion (TG) of infected/immunized pigs. Pigs were infected with PRV wild-type (wt) or immunized with PRVtmv+ vaccine. At 28 days post-infection/immunization, latently infected pigs were reactivated by Dex injection. At 5 days post-reactivation, pigs were euthanized, and TG was collected. Total RNA was isolated from 25 mg of TG tissues. DNase-treated RNA was used for cDNA synthesis, and qPCR was performed subsequently targeting PRV (**A**) immediate early gene transcript ICP0 and late gene transcripts (**B**) major capsid protein (MCP) and (**C**) glycoprotein (gC). DNase-treated RNA without cDNA synthesis was included as a control to determine the efficacy of DNase treatment. Targeted transcript copies were calculated according to CT values of the standard curve. Calculated transcript copies were normalized per ng of RNA. Two independent qPCR analyses were performed for each animal. The bar graph represents the individual values in each group. PRV wt—PRV wild-type virus-infected pigs (*n* = 2); PRVtmv+—PRVtmv+ vaccine-immunized pigs, latency reactivated (*n* = 3); Negative—TGs from mock-infected pigs (stored in −80 °C) from the previous study [1].

**Figure 7 viruses-15-00473-f007:**
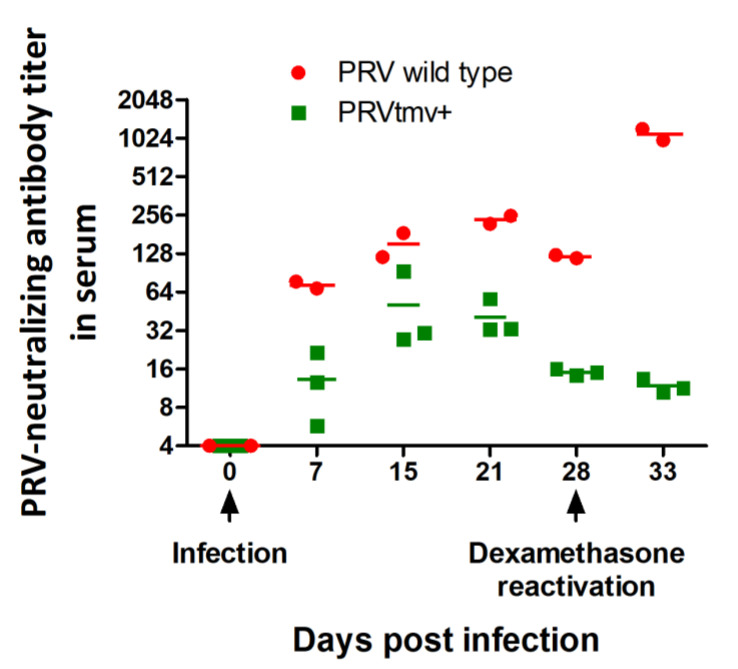
Pseudorabies virus (PRV)-specific serum neutralizing antibody titer developed in pigs following infection. The data represent the mean ± standard deviation. (*n* = 2 for PRV wild-type group and *n* = 3 for PRVtmv+ group.)

**Table 1 viruses-15-00473-t001:** Clinical scoring scheme.

Clinical Scoring Criteria	Rectal Temperature (°C)	Nasal Discharge	Lethargy	Dyspnea	Anorexia	Cough
Normal (0)	<39.7	None	Normal in attitude	Normal breathing	Normal appetite	None
Mild (1)	39.8–40.3	Serous	Moves slowly, head down	Slight difficulty breathing	Slightly off feed	<3 occasional cough
Moderate (2)	40.4–40.8	Mucopurulent	Tends to lie down, moves reluctantly	Labored breathing	Moderately off feed	>3 repeated cough
Severe (3)	40.9–41.1	Copious mucopurulent	Difficult to stand, little response to stimuli	Very labored breathing	Not eating	NA
(4)	>41.2	NA	NA	NA	NA	NA

NA—not applicable.

**Table 2 viruses-15-00473-t002:** List of primers, probes, and double-stranded gene blocks (ds-gblock as standard) used in quantitative PCR for quantification of pseudorabies (PRV) ICP0, major capsid protein, and glycoprotein C gene/transcript copies in tissue samples and subsequent normalization based on glyceraldehyde 3-phosphate dehydrogenase (GAPDH) housekeeping gene.

Primer/Probe/ds-Gblock	Name	Sequence
PRV-ICP0	Forward	5′-atcccgtgctcctggataatctcg-3′
	Reverse	5′-tccccgtcttcaactggctttatg-3′
	Probe	5′Fam-atgttgtccacgacggcctcgcgga-3′ Tamra
	ds-gblock	5′-ggcctcggtcacgcgctggcggttcatcccgtgctcctggataatctcgacg Atgttgtccacgacggcctcgcggatggggtcgctctcgatgaccgtcgagacct gcccataaagccagttgaagacggggactctggggcgggcgcgagacccaga-3′
PRV–Major Capsid Protein	Forward	5′-ccatccagtttgaggtgcag-3′
	Reverse	5′-cgaggcgcttgatcatgtag-3′
	Probe	5′Fam-cccgtcgcgcgcgatcatcg-3′ Tamra
	ds-gblock	5′-ctcagctacgtggccgagggcaccatccagtttgaggtgcagcagccgatg atcgcgcgcgacgggccgcacccggccgaccagcccgtgcacaactacatgatca agc gcctcgatcgccgctccctcaacgccgc-3′
PRV–Glycoprotein C	Forward	5′-gtcgtccgcgactactacc-3′
	Reverse	5′-tcacgttcaccacggagac-3′
	Probe	5′Fam-cgtccgcgaaccagcgcagg-3′ Tamra
	ds-gblock	5′-agcccttccgggcggtgtgcgtcgtccgcgactactacccgcggcgcagcgtg cgcctgcgctggttcgcggacgagcacccggtggacgccgccttcgtgaccaa cagcaccgtggccgacgagctcgggcgccgcacgcgcgtctccgtggtgaac gtgacgcgcgcggacgtcccgggc-3′
Swine Glyceraldehyde 3-Phosphate Dehydrogenase (GAPDH)	Forward	5′-atgacaacttcggcatcgtg-3′
	Reverse	5′-ccatccacagtcttctgggt-3′
	Probe	5′Fam-accacagtccatgccatcactgcc-3′ Tamra
	ds-gblock	5′-gcacccctggccaaggtcatccatgacaacttcggcatcgtggaaggactca tgaccacagtccatgccatcactgccacccagaagactgtggatggc ccctctgggaaacgtggcgt-3′

**Table 3 viruses-15-00473-t003:** Nasal shedding following infection determined by pseudorabies virus-specific qPCR and virus isolation in swine kidney (SK) cells.

Group	Animal #	Virus Isolation (PFU/mL)		PRV Genome Copies per 100 ng of DNA (Days Post-Infection)				
		Day 3	Day 5	Day 3	Day 5	Day 15	Day 21	Day 28
PRV wild-type	2303	4.0 × 10^4^	0	3.1 × 10^6^	1.9 × 10^4^	5.7 × 10^2^	2.0 × 10^2^	0
	2306	7.6 × 10^4^	0	6.7 × 10^6^	4.1 × 10^4^	3.6 × 10^2^	3.8 × 10^2^	0
PRVtmv+	2309	0	0	2.9 × 10^1^	5.2 × 10^1^	0	0	0
	2312	0	0	2.4 × 10^2^	1.4 × 10^2^	0	0	0
	2315	0	0	8.1 × 10^1^	1.8 × 10^2^	0	0	0

## Data Availability

Not applicable.

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
