# Peer review of "Live Triple Gene-Deleted Pseudorabies Virus-Vectored Subunit PCV2b and CSFV Vaccine Undergoes an Abortive Replication Cycle in the TG Neurons following Latency Reactivation"

_viruses, 2023, doi:10.3390/v15020473_

Round 1

Reviewer 1 Report

The authors reported that PRV-tmv+, a chimered virus that three genes TK/ gE/ gG were replaced by cap of PCV2b, E2 of CSFV and Erns-GM-CSF, infected TG neurons and could not replication, suggesting that PRV-tmv (vector) is safe. If the authors only test vector, they should test PRV-tmv without three inserted genes. If they only test the chimeric vaccines safe, the should test PRV-tmv+ with the E2, Cap and other gene expression, and test whether the inserted genes affect PRVs replication. Therefore, all studies should contains PRV, PRV-tmv and PRV-tmv+ three groups, not two groups.  

Specific points:

1. There are too few experimental animals in PRV-wt and PRV-tmv+ groups, the experimental results are not very convincing.

2. Whether injecting antibiotics into pigs will have an effect on the activation test.

3. At least three data are needed for statistical analysis, while the PRV-wt group has only two pigs?

4. Some experiments should be added to confirm PRV-tmv+ infection, including WB, PCR, IFA results showing E2 , Cap expression.

5. The author should compare PRV-wt, PRV-tmv, and PRV-tmv+ in this study.

6. The author should test whether PRV-WT and PRVtmv+ infects and replication in TG. qPCR is OK, but it is not enough. I suggest the authors test whether the TG is infected by using IFA (test gene expression ) and RNAscope (label viral RNA) and FISH (for viral DNA).

7. PRV lacking TK fails to replicate in the trigeminal nerve. PRV-tmv+ should fail to replication and PRV-tmv+ is not activated by dexamethasone? Is it right.

8. PRV-tmv+ is not activated by dexamethasone, the authors did not test whether the inserted three genes inserted affect PRV infection in TG neurons.

9. According to the title, the author should test PRV-tmv (vector), but not PRV-tmv+ (chimeric virus). In fact, the author only test whether the chimeric virus infects TG neurons and replicate or not and whether the chimeric vaccine is safe. So, the title is not right, Please change it.

10. To report your result: However should be: we noticed that ....,; Consequently should be In addition, we found that....; Therefore should be : Our results demonstrated that....

Author Response

The authors reported that PRV-tmv+, a chimered virus that three genes TK/ gE/ gG were replaced by cap of PCV2b, E2 of CSFV and Erns-GM-CSF, infected TG neurons and could not replication, suggesting that PRV-tmv (vector) is safe. If the authors only test vector, they should test PRV-tmv without three inserted genes. If they only test the chimeric vaccine’s safe, the should test PRV-tmv+ with the E2, Cap and other gene expression and test whether the inserted genes affect PRV’s replication. Therefore, all studies should contains PRV, PRV-tmv and PRV-tmv+ three groups, not two groups. 

Specific points:

  1. Comment: There are too few experimental animals in PRV-wt and PRV-tmv+ groups, the experimental results are not very convincing.

Response: We understand the reviewer’s concern regarding using two animals in the PRV wildtype (WT) group for the latency-reactivation study. It is a well-established and documented knowledge that all animal alpha herpesviruses establish latency in the sensory neurons of the trigeminal ganglia upon primary intranasal infection. Further, it is also well-accepted knowledge that envelope proteins gE, gG, and TK genes are not necessary for virus entry into the nerve termini of the trigeminal nerve and retrograde transport to the cell bodies in the TG neurons where both WT and PRVtmv+ (gE, gG and TK-deleted) viruses establish latency  (Arch Virol (1994) 136:197-205 [1]; Vet Micro (1998) 62(3): 171-183[2]; Virology (1986), 155: 600-613 [3]; Journal of Wildlife Diseases, (2003) 39(3): 567–575 [4]; The Journal of infectious diseases (2002) 186 [5]). In vitro,  in cell culture, linear viral genome circularizes prior to a well regulated gene transcription/translation scheme. Immediate early and early genes are transcribed/translated  first before the genome replication and late gene transcription/translation. During latency, the viral genome assumes a circular configuration and resides in the nucleus of the neuron. Further,  with the exception of latency-associated transcript (LAT) there is no other gene transcription or translation and no productive virus replication during latency.   Upon stress or dexamethasone-induced reactivation of the latent herpesvirus, the latent virus reactivates and shed in the nasal secretion [6]. In the TG neurons, viral gene transcription is also similarly regulated as in cell culture; the immediate early gene transcription occurs first from the latent circular genome upon reactivation. This is followed by early gene transcription and translation, which includes virally coded TK, required for the virus replication. Lastly, late proteins are produced, including capsid and envelope proteins required for virus assembly. Notably, many, if not all, late protein transcription and translation are dependent on DNA replication.

We designed the PRVtmv with the following hypothesis; i) it can be used as a live subunit vaccine vector for immunization against other swine viruses because the virus will replicate in the nasal epithelium and induce mucosal, humoral and cellular immune responses, ii) Intranasal vaccination route with live virus vector would be less affected by preexisting immune interference; iii) After primary infection of the nasal epithelium,  the virus would enter the nerve endings of the trigeminal nerve and be transported to the cell bodies in the TG neurons and iv) the vaccine virus will establish latency in the TG neurons, but under no circumstances the virus can be shed in the nasal secretions; Therefore,  the vaccine virus will not be transmitted to other pigs or maintained in the pig population and v) Finally, the vaccinated animals can be distinguished from the PRV wt infected pigs  based on its DIVA (distinguishable from the field wildtype PRV) property, which will be critical if the vaccine needs to be used in PRV-free countries. We have already characterized PRVtmv+ in vitro and in vivo in pigs, and determined its vaccine efficacy against PCV2b (Vaccines, 2022 Feb 16;10(2):305 [7]); and CSFV (Chowdhury et al., 2023 – manuscript in preparation [8]).

Our experimental design for this study was based on the following dogmas: i) Neurons are postmitotic and, as such, do not code for TK enzyme; therefore, the TK minus viruses do not replicate in the neurons (Arch Virol (1991) 120:57-70 [9]),  ii) TK minus animal alpha herpesviruses replicate in vitro in nonneuronal epithelial cells and in vivo in nasal or other epithelial cells, because the cellular TK in these cells can complement for the viral TK; therefore, they can be readily propagated in cell cultures, the stock viruses can be prepared, and they are suitable as a live vaccine. iii) alpha herpesvirus anterograde axonal neuronal transport from the neuronal cell body to the axon termini is mediated by gE and Us9 and that axonal neuronal transport of infectious fully enveloped PRV occurs within transport vesicles along the axon [10,11]. Therefore, virus replication and assembly in the TG neurons must first be completed before the virus can be transported to the nasal epithelium and shed in the nasal secretions. We hypothesized that regardless of PRV wt or PRVtmv+, both viruses would reactivate upon Dex and transcribe the immediate early gene ICP0; however, only the PRV wt and not the PRVtmv+ would transcribe the early protein TK, required for virus replication in the TG neurons. As a result, the PRV wt and not the PRVtmv+ can replicate in TG neurons and forms the enveloped virus particles for vesicular anterograde transport, to the nerve termini in the nasal epithelium and the nasal virus shedding.

Based on the above discussion,  we never questioned the establishment of latency in the TG’s of both PRV wt and PRVtmv+ after primary intranasal infection. Also, for PRV wt, Dex-induced latent virus reactivation and nasal virus shedding were expected. Thus we used only two pigs as a positive control for PRV wt. For PRV tmv+, we expected the virus not to replicate in the TG neurons, even if the latent virus was reactivated. To prove that the PRVtmv+ reactivated in the TGs of Dex-treated pigs, we tested the transcription of the immediate early gene, ICP0, which would be transcribed immediately after reactivation regardless of whether the virus expressed TK or not. Our results demonstrated the ICP0 transcription in the  TG’s of all three Dex-treated PRVtmv+ infected pigs. To prove that PRVtmv+ failed to replicate in the same TG neurons, we compared the number of PRVtmv+ genomic copies between the TG’s of latently infected- (from our previous experiment; [7]) and the Dex-treated pigs. As expected, the genomic copy numbers in both cases were very similar. The final proof that PRVtmv+ did not replicate in the TG neurons of Dex-treated pigs was that the DNA replication-dependent gamma 2 (late gene) gC transcription was not detectable in all three PRVtmv+ - infected pigs. Also, the leaky late (partially DNA replication-dependent),  major capsid protein VP5 gene was not transcribed. On the contrary, both gC and VP5 transcripts were readily detectable in the TG’s of Dex-treated wt-infected pigs (positive control). We have added a new section in the discussion (Line number 343 - 423)

  1. Comment: Whether injecting antibiotics into pigs will have an effect on the activation test?

Response: Antibiotics do not have any role in reactivation from latency (in both in vitro and in vivo) in any known animal and human herpesvirus, including the pseudorabies virus. The purpose of injecting antibiotic (Noromycin® 300 LA; Oxytetracycline) at the time of experimental virus/vaccine inoculation was to prevent secondary bacterial infections. So, it is not expected to have any effect on latency-reactivation.

  1. Comment: At least three data are needed for statistical analysis, while the PRV-wt group has only two pigs?

Response: We have addressed this question in answer to comment #1. (Line number 343 - 423)

  1. Comment: Some experiments should be added to confirm PRV-tmv+ infection, including WB, PCR, IFA results showing E2, Cap expression.

Response: Characterization of the PRVtmv+ for the expression of the chimeric proteins by WB and IFA has been reported earlier [7]. As given in the methods section and results, we performed PRV-specific qPCR to determine the infection/establishment of latency in the trigeminal ganglionic neurons (both in PRV wildtype and PRVtmv+ vaccine group). Performing PCR/qPCR to demonstrate PRV DNA in TG is sufficient to confirm the infection/establishment of latency in the TG neurons in herpes virology (Arch Virol (1994) 136:197 205 [1]; Journal of Wildlife Diseases, (2003) 39(3): 567–575 [4]; Veterinary Microbiology, 24 (1990) 281-295 [12]). Subsequently, we performed PRV-specific quantification of mRNA transcript targeting i) immediate early gene protein (ICP0) and ii) late proteins (Major capsid protein gene [MCP] and glycoprotein C), which demonstrates reactivation from latency (both wt and PRVtmv+),  virus replication (only for wt), respectively. So, presence or absence of CSFV and PCV2b chimeric proteins expression in the trigeminal ganglionic neurons does not have any role and significance in the latency-reactivation property of PRVtmv+ vaccine virus in pigs.

  1. Comment: The author should compare PRV-wt, PRV-tmv, and PRV-tmv+ in this study.

Response: We agree with the reviewer’s comments regarding including the PRV-tmv vector without and chimeric insert for the infection studies. We recently characterized our PRVtmv+ vaccine virus against PCV2b and CSFV challenge in pigs (Vaccines (Basel). 2022 Feb 16;10(2):305 [7]; Chowdhury et al., 2023 – manuscript in preparation [8]). As a part of further characterization and safety of our vaccine in pigs, our goal for the present study is only to investigate the latency-reactivation property of our live triple gene-deleted pseudorabies virus vectored subunit PCV2b and CSFV vaccine (PRVtmv+). Testing the latency-reactivation property of our bivalent CSFV and PCV2b vaccine is very important from safety point of view. Because it determines the vaccine virus persistence in the immunized pig population.

  1. Comment: The author should test whether PRV-WT and PRVtmv+ infects and replication in TG. qPCR is OK, but it is not enough. I suggest the authors test whether the TG is infected by using IFA (test gene expression) and RNAscope (label viral RNA) and FISH (for viral DNA).

Response: As we mentioned in our response to the comment 4, we performed PRV-specific qPCR to demonstrate PRV DNA TG neurons to demonstrate the infection/establishment of latency in the trigeminal ganglionic neurons (both in PRV wildtype and PRVtmv+ vaccine group). Reactivation assay and subsequent qPCR for PRV-specific mRNA transcripts (targeting i) immediate early gene protein (ICP0) and ii) late proteins (Major capsid protein gene [MCP] and glycoprotein C) showed our vaccine did not replicate and shed through nostril. Fluorescence in situ hybridization (FISH)/RNAscope to locate the viral DNA/RNA with in the trigeminal ganglionic neurons may again confirms our findings of abortive PRVtmv+ replication in a different way. However, it is not in the scope for our current study.

  1. Comment: PRV lacking TK fails to replicate in the trigeminal nerve. PRV-tmv+ should fail to replication and PRV-tmv+ is not activated by dexamethasone? Is it right.

Response: Yes. As the reviewer correctly mentioned, the PRV lacking thymidine kinase does not replicate in the trigeminal ganglionic neurons. Deleting viral thymidine kinase from the PRV (PRVtmv+) and absence of cellular thymidine kinase in trigeminal ganglionic neurons, PRV replication following dexamethasone-induced latency-reactivation is inhibited (Line number 287-299).

  1. Comment: PRV-tmv+ is not activated by dexamethasone, the authors did not test whether the inserted three genes inserted affect PRV infection in TG neurons.

Response: The chimeric genes inserted into the PRVtmv genomes to construct the PRVtmv+ vaccine are i) Capsid protein from porcine circovirus type 2b (PCV2b), envelope proteins ii) E2 and iii) Erns (fused with porcine Granulocyte-macrophage colony-stimulating factor [GM-CSF]) from classical swine fever virus (CSFV). As the inserted chimeric proteins are from unrelated virus families (PCV2b – family Circoviridae and CSFV – family Flaviviridae) to that of pseudorabies virus herpesvirus (family Herpesviridae), we are not expected to see any effects of these chimeric protein in the latency-reactivation following the dexamethasone injection in PRVtmv+ immunized pigs. Therefore, we have addressed the concern in our response to the comment #4.

  1. Comment: According to the title, the author should test PRV-tmv(vector), but not PRV-tmv+ (chimeric virus). In fact, the author only test whether the chimeric virus infects TG neurons and replicate or not and whether the chimeric vaccine is safe. So, the title is not right, Please change it.

Response: We completely agree with the reviewer’s comment. The original title was “Live Triple Gene-Deleted Pseudorabies Virus Vaccine Vector undergoes an abortive replication cycle in the TG neurons following latency-reactivation”. We have revised the title to “Live Triple Gene-Deleted Pseudorabies Virus- Vectored Subunit PCV2b and CSFV Vaccine undergoes an abortive replication cycle in the TG neurons following latency-reactivation” (Line number: 2-4)

  1. Comment: To report your result: However should be: we noticed that ....,; Consequently should be In addition, we found that....; Therefore should be : Our results demonstrated that...

Response: We have revised the sentences as suggested. Line numbers: 23-27, 71-73, 190 and 284-285.

Reviewer 2 Report

In the current manuscript, “Live Triple Gene-Deleted pseudorabies Virus Vaccine Vector undergoes an abortive replication cycle in the TG neurons following latency-reactivation” Authors have investigated the in vivo replication, latency reactivation, and shedding in the nasal secretion of PRVtmv+ viral vaccine vector. Below I have mentioned a few of my concerns related to the current manuscript.

“The viral Thymidine Kinase (TK) is essential for viral replication in the trigeminal ganglionic (TG) neurons” this information is self-explanatory that deletion mutant (PRVtmv+) of this gene most likely will fail to replicate in the TGs.

Concern#1: What is the prime significance of using PRVtmv+ as a vaccine vector? How is this vector better than a double mutant that only lacks gG and gE? And why it is essential to study the latency reactivation of this triple mutant virus.

I would suggest that authors should consider providing this explanation, which will help readers to understand the significance of this study.

In their previous published study (https://doi.org/10.3390/vaccines10020305), authors reported similar in vitro growth kinetics, where no difference in the replication was observed between PRV wt and PRVtmv+ viruses in the swine kidney cells.

Concern#2: PRVtmv+ is a replication-competent virus in epithelial cells (in the presence of cellular TK). However, in the current study, no infectious virus particle was isolated in nasal shedding even though a higher infection dose was used compared to the wt virus group. Authors should comment on this observation.

Concern#3: What is the rationale behind the use of subcutaneous injection?

Concern#4: Where is the number of pigs used sufficiently to provide statistical significance? 

Author Response

In the current manuscript, “Live Triple Gene-Deleted pseudorabies Virus Vaccine Vector undergoes an abortive replication cycle in the TG neurons following latency-reactivation” Authors have investigated the in vivo replication, latency reactivation, and shedding in the nasal secretion of PRVtmv+ viral vaccine vector. Below I have mentioned a few of my concerns related to the current manuscript.

“The viral Thymidine Kinase (TK) is essential for viral replication in the trigeminal ganglionic (TG) neurons” this information is self-explanatory that deletion mutant (PRVtmv+) of this gene most likely will fail to replicate in the TGs.

  1. Comment: a) What is the prime significance of using PRVtmv+ as a vaccine vector? b) How is this vector better than a double mutant that only lacks gG and gE? c) And why it is essential to study the latency reactivation of this triple mutant virus.

I would suggest that authors should consider providing this explanation, which will help readers to understand the significance of this study.

Response to a: We agree with the reviewer’s comments. a) The advantage of using PRVtmv as vaccine vector are as follows: i) The PRVtmv+ is highly attenuated, yet replicates well in the nasal mucosa, ii) highly immunogenic, therefore induced strong immune response in pigs, ii) has DIVA property iii) Can potentially accommodate large inserts – 10-15 kb. iv) Stably express inserted chimeric proteins and induced strong protective immune response, v) Vector virus establishes latency in trigeminal ganglion  and reactivates  but failed to replicate in the TG neurons and therefore no nasal virus shedding and vi) Suitable for construction of polyvalent subunit vaccines. We have added these advantages in our revised version of the manuscript (Line number 453-460).

Response to b: Regarding why we additionally deleted the TK gene beside the gE and gG deletions the rationale is as follows:  The objective for the TK deletion are i) we wanted to make sure our vaccine virus do not replicate in the TG neurons following reactivation which in turn will prevent nasal virus shedding, ii) It creates the additional space for the chimeric genes insertion. Further details also provided in the response to the reviewer # 1, comment #1 (Line number 355 - 423).  

Response to c: One of our hypotheses for the PRVtmv+ design was that  i) after primary infection of the nasal epithelium,  the virus would enter the nerve endings of the trigeminal nerve and be transported to the cell bodies in the TG neurons and ii) the vaccine virus will establish latency in the TG neurons, but after latency-reactivation the virus cannot be shed in the nasal secretions. Our results in this study proved our hypothesis. Therefore, our vaccine virus would not pose any threat for circulation in the pig population. Consequently, the vaccine could be introduced in PRV free countries. 

  1. Comment: In their previous published study (https://doi.org/10.3390/vaccines10020305), authors reported similar in vitro growth kinetics, where no difference in the replication was observed between PRV wt and PRVtmv+ viruses in the swine kidney cells. PRVtmv+ is a replication-competent virus in epithelial cells (in the presence of cellular TK). However, in the current study, no infectious virus particle was isolated in nasal shedding even though a higher infection dose was used compared to the wt virus group. Authors should comment on this observation.

Response: In our previous study [7] we used 10 fold more PRVtmv+ for the intranasal infection and we could isolate lower amount of infectious virus from the nasal swabs. It is a well-established knowledge that even though the gE-deleted virus replicates like the wt in vitro in epithelial cells, it replicates with a much lower titers in vivo in the nasal epithelium. Therefore, more sensitive methods such as qPCR are preferable.

  1. Comment: What is the rationale behind the use of subcutaneous injection?

Response: The pigs in the PRVtmv+ vaccine group were inoculated intranasally with 8 × 106 PFUs and subcutaneously with 4 × 106 PFUs per pig. Mucosal administration (intranasal) stimulates higher levels of mucosal IgA antibodies that can inhibit disease transmission with greater effectiveness. However, intranasal inoculation induces less humoral immune response. Inoculation through subcutaneous causes slower absorption thus, it is a good route if a prolonged effect is to be achieved. Subcutaneous administration has emerged as the proposed route of administration for several new vaccines under development, with the intention to increase immunogenicity. The subcutaneous route induced a higher level of neutralizing antibodies than the intradermal and intramuscular vaccination routes (Front. Immunol. (2021) 12:645210 [13]). Immune response toward subcutaneously administered antigens likely entails two waves of antigen presentation by both migratory skin-resident and lymph node-resident dendritic cells, which likely drive better immunogenicity (BioDrugs volume (2021) 35, pages125–146 [14]). For optimal immune stimulation we have used both intranasal and subcutaneous route to administer our PRVtmv+ vaccine in pigs.

  1. Comment: Where is the number of pigs used sufficiently to provide statistical significance?

Response: As we addressed the similar concern raised by the reviewer 1 in response to the reviewer # 1, comment #1 (Line number: 355-423)

Round 2

Reviewer 1 Report

    The authors had answered all questions I concerned.